# An ancestral interaction module promotes oligomerization in divergent mitochondrial ATP synthases

Ondřej Gahura [1,4], Alexander Mühleip [2,4], Carolina Hierro-Yap[1,3], Brian Panicucci[1], Minal Jain[1,3], David Hollaus [3], Martina Slapničková[1], Alena Zíková [1,3] ✉ & Alexey Amunts [2] ✉

Mitochondrial ATP synthase forms stable dimers arranged into oligomeric assemblies that generate the inner-membrane curvature essential for efficient energy conversion. Here, we report cryo-EM structures of the intact ATP synthase dimer from *Trypanosoma brucei* in ten different rotational states. The model consists of 25 subunits, including nine lineage-specific, as well as 36 lipids. The rotary mechanism is influenced by the divergent peripheral stalk, conferring a greater conformational flexibility. Proton transfer in the lumenal half-channel occurs via a chain of five ordered water molecules. The dimerization interface is formed by subunit-*g* that is critical for interactions but not for the catalytic activity. Although overall dimer architecture varies among eukaryotes, we find that subunit-*g* together with subunit-*e* form an ancestral oligomerization motif, which is shared between the trypanosomal and mammalian lineages. Therefore, our data defines the subunit-*g/e* module as a structural component determining ATP synthase oligomeric assemblies.

Mitochondrial ATP synthase consists of the soluble $F_1$ and membrane-bound $F_o$ subcomplexes and occurs in dimers that assemble into oligomers to induce the formation of inner-membrane folds, called cristae. The cristae are the sites for oxidative phosphorylation and energy conversion in eukaryotic cells. Dissociation of ATP synthase dimers into monomers results in the loss of native cristae architecture and impairs mitochondrial function[1,2]. While cristae morphology varies substantially between organisms from different lineages, ranging from flat lamellar in opisthokonts to coiled tubular in ciliates and discoidal in euglenozoans[3], the mitochondrial ATP synthase dimers represent a universal occurrence to maintain the membrane shape[4].

ATP synthase dimers of variable size and architecture, classified into types I to IV have recently been resolved by high-resolution cryo-EM studies. In the structure of the type-I ATP synthase dimer from mammals, the monomers are only weakly associated[5,6], and in yeast insertions in the membrane subunits form tighter contacts[7]. The structure of the type-II ATP synthase dimer from the alga *Polytomella*

sp. showed that the dimer interface is formed by phylum-specific components[8]. The type-III ATP synthase dimer from the ciliate *Tetrahymena thermophila* is characterized by parallel rotary axes, and a substoichiometric subunit, as well as multiple lipids were identified at the dimer interface, while additional protein components that tie the monomers together are distributed between the matrix, transmembrane, and lumenal regions[9]. The structure of the type-IV ATP synthase with native lipids from *Euglena gracilis* also showed that specific protein-lipid interactions contribute to the dimerization, and that the central and peripheral stalks interact with each other directly[10]. Finally, a unique apicomplexan ATP synthase dimerizes via 11 parasite-specific components that contribute ~7000 Å$^2$ buried surface area[11], and unlike all other ATP synthases, that assemble into rows, it associates in higher oligomeric states of pentagonal pyramids in the curved apical membrane regions. Together, the available structural data suggest a diversity of oligomerization, and it remains unknown whether common elements mediating these interactions exist or whether

[1]Institute of Parasitology, Biology Centre, Czech Academy of Sciences, 37005 České Budějovice, Czech Republic. [2]Science for Life Laboratory, Department of Biochemistry and Biophysics, Stockholm University, 17165 Solna, Sweden. [3]Faculty of Science, University of South Bohemia, 37005 České Budějovice, Czech Republic. [4]These authors contributed equally: Ondřej Gahura, Alexander Mühleip. ✉e-mail: azikova@paru.cas.cz; amunts@scilifelab.se

dimerization of ATP synthase occurred independently and multiple times in evolution[4].

The ATP synthase of *Trypanosoma brucei*, a representative of kinetoplastids and an established medically important model organism causing the sleeping sickness, is highly divergent, exemplified by the pyramid-shaped $F_1$ head containing a phylum specific subunit[12,13]. The dimers are sensitive to the lack of cardiolipin[14] and form short left-handed helical segments that extend across the membrane ridge of the discoidal cristae[15]. Uniquely among aerobic eukaryotes, the mammalian life cycle stage of *T. brucei* utilizes the reverse mode of ATP synthase, using the enzyme as a proton pump to maintain mitochondrial membrane potential at the expense of ATP[16,17]. In contrast, the insect stages of the parasite employ the ATP-producing forward mode of the enzyme[18,19].

Given the conservation of the core subunits, the different nature of oligomerization and the ability to test structural hypotheses biochemically, we reasoned that investigation of the *T. brucei* ATP synthase structure and function would provide the missing evolutionary link to understand how the monomers interact to form physiological dimers.

Here, we address this question by combining structural, functional, and evolutionary analysis of the *T. brucei* ATP synthase dimer.

## Results

### Cryo-EM structure of the *T. brucei* ATP synthase

We purified ATP synthase dimers from cultured *T. brucei* procyclic trypomastigotes by affinity chromatography with a recombinant natural protein inhibitor TbIF$_1$[20] and subjected the sample to cryo-EM analysis (Supplementary Figs. 1 and 2). Using masked refinements, maps were obtained for the membrane region, the rotor, and the peripheral stalk. To describe the conformational space of the *T. brucei* ATP synthase, we resolved ten distinct rotary substates, which were refined to 3.5–4.3 Å resolution. Finally, particles with both monomers in rotational state 1 were selected, and the consensus structure of the dimer was refined to 3.2 Å resolution (Supplementary Table 1, Supplementary Figs. 2 and 3).

Unlike the wide-angle architecture of dimers found in animals and fungi, the *T. brucei* ATP synthase displays an angle of 60° between the two $F_1/c$-ring subcomplexes. The model of the *T. brucei* ATP synthase includes all 25 different subunits, nine of which are lineage-specific (Fig. 1a, Supplementary Fig. 4, and Supplementary Movie 1). We named the subunits according to the previously proposed nomenclature[21–23] (Supplementary Table 2). In addition, we identified and modelled 36 bound phospholipids, including 24 cardiolipins (Supplementary Fig. 5). Both detergents used during purification, n-dodecyl β-D-

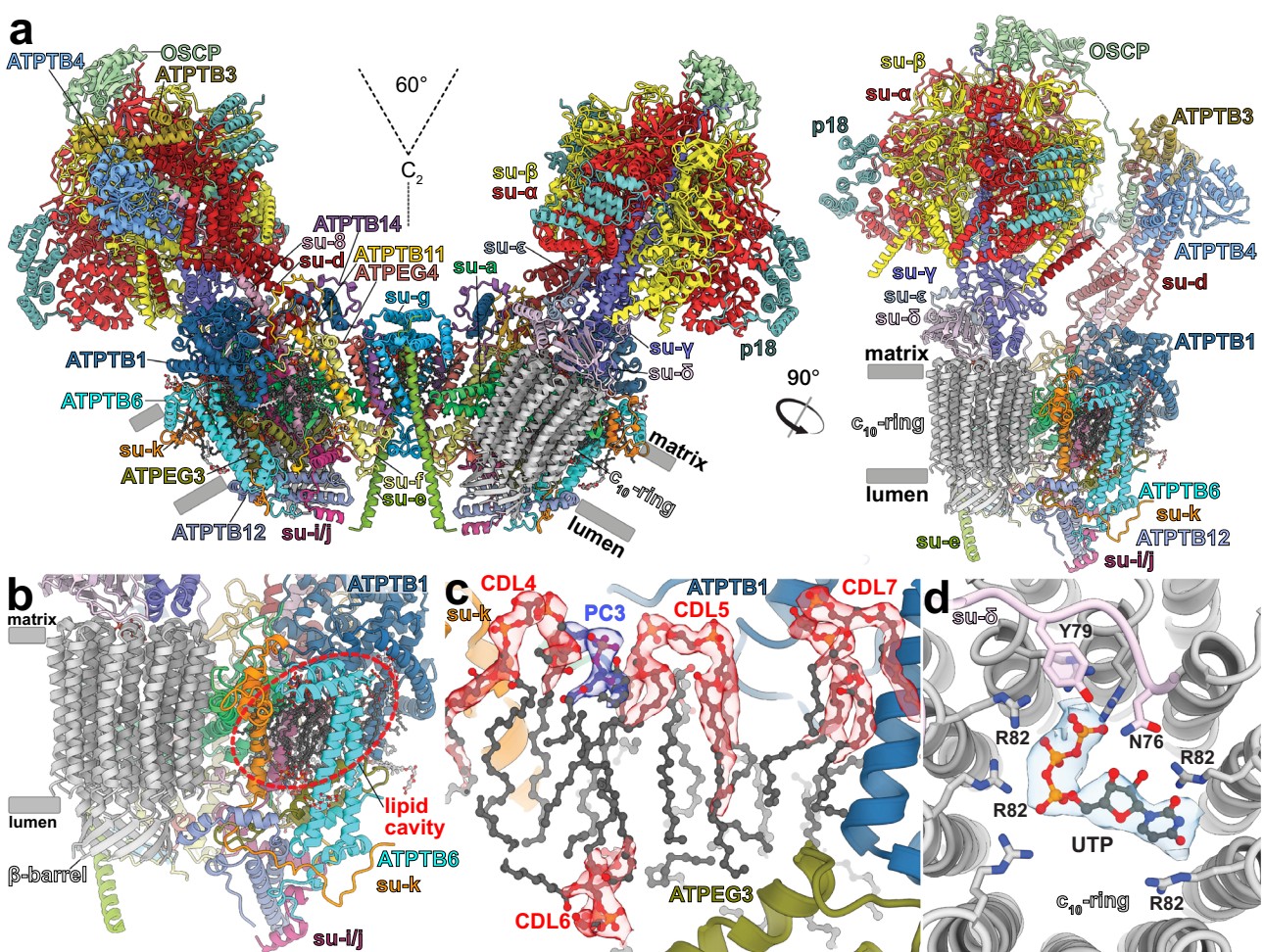

**Fig. 1 | The *T. brucei* ATP synthase structure with lipids and ligands. a** Front and side views of the composite model with both monomers in rotational state 1. The two $F_1/c_{10}$-ring complexes, each augmented by three copies of the phylum-specific p18 subunit, are tied together at a 60°-angle. The membrane-bound $F_o$ region displays a unique architecture and is composed of both conserved and phylum-specific subunits. **b** Side view of the $F_o$ region showing the lumenal interaction of the ten-stranded β-barrel of the *c*-ring (grey) with ATPTB12 (pale blue). The lipid-filled peripheral $F_o$ cavity is indicated. **c** Close-up view of the bound lipids within the peripheral $F_o$ cavity with cryo-EM density shown. **d** Top view into the decameric *c*-ring with a bound pyrimidine ribonucleoside triphosphate, assigned as UTP, although not experimentally detected. Map density shown in transparent blue, interacting residues shown.

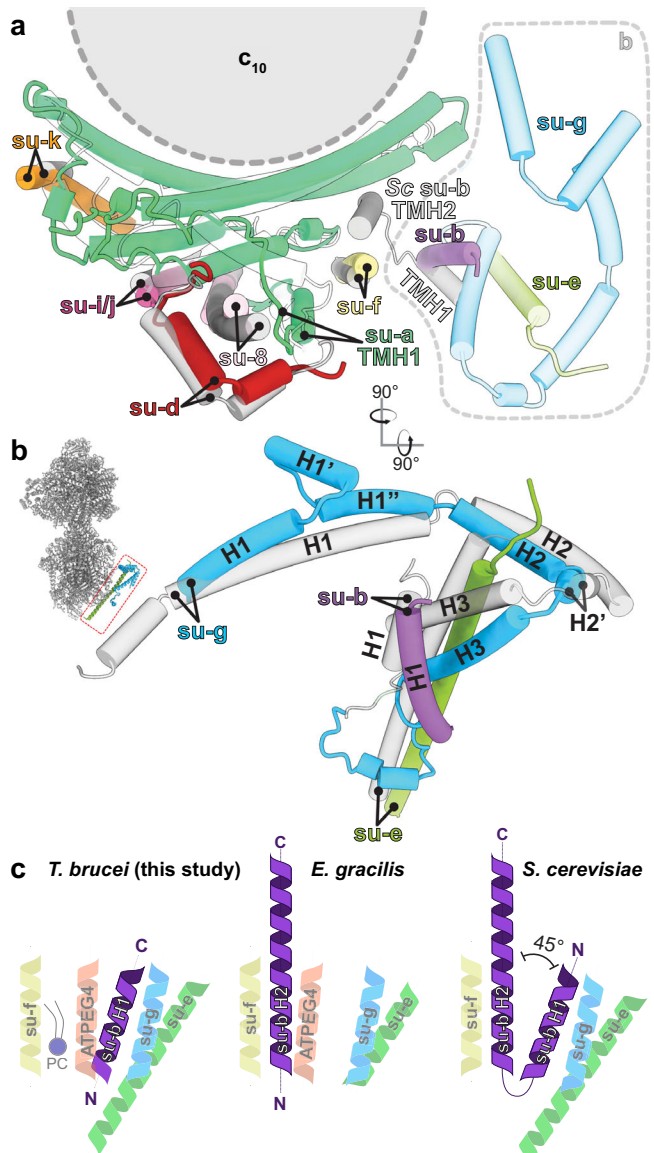

**Fig. 2 | Identification of conserved F_o subunits. a** Top view of the membrane region with *T. brucei* subunits (coloured) overlaid with *S. cerevisiae* structure (grey transparent). Close structural superposition and matching topology allowed the assignment of conserved subunits based on matching topology and location. **b** Superposition of subunits-*b*, -*e* and -*g* with their *S. cerevisiae* counterparts PDB 6B2Z (*S. cerevisiae* mitochondrial ATP synthase) confirms their identity. **c** Schematic representation of transmembrane helices of subunit-*b* and adjacent subunits in *T. brucei*, *E. gracilis* PDB 6TDV (*E. gracilis* mitochondrial ATP synthase, membrane region)[10] and *S. cerevisiae* PDB 6B2Z (*S. cerevisiae* mitochondrial ATP synthase)[7] ATP synthases. PC – phosphatidylcholine.

maltoside (β-DDM) and glyco-diosgenin (GDN) are also resolved in the periphery of the membrane region (Supplementary Fig. 6).

In the catalytic region, F_1 is augmented by three copies of subunit p18, each bound to subunit-*α*[12,13]. Our structure shows that p18 is involved in the unusual attachment of F_1 to the peripheral stalk. The membrane region includes eight conserved F_o subunits (*b*, *d*, *f*, 8, *i/j*, *k*, *e*, and *g*) arranged around the central proton translocator subunit-*a*. We identified those subunits based on the structural similarity and matching topology to their yeast counterparts (Fig. 2). For subunit-*b*, a single transmembrane helix superimposes well with *b*H1 from yeast and anchors the subunits-*e* and -*g* to the F_o (Fig. 2a, b). In yeast and bovine ATP synthases *b*H1 and transmembrane helices of subunits-*e*

and -*g* are arranged in the same way as in our structure and contribute to a characteristic wedge in the membrane domain[5]. The long helix *b*H2, which constitutes the central part of the peripheral stalk in other organisms is absent in *T. brucei* (Fig. 2c). No alternative subunit-*b*[24] is found in our structure.

The membrane region contains a peripheral subcomplex, formed primarily by the phylum-specific ATPTB1,6,12 and ATPEG3 (Fig. 1b). It is separated from the conserved core by a membrane-intrinsic cavity, in which nine bound cardiolipins are resolved (Fig. 1c), and the C-terminus of ATPTB12 interacts with the lumenal β-barrel of the *c*_10-ring. The β-barrel, which has previously been reported also in the ATP synthase from *E. gracilis*[10], extends from the *c*_10-ring approximately 15 Å to the lumen (Fig. 1a and Supplementary Fig. 7). The cavity of the decameric *c*-ring contains density consistent with disordered lipids, as observed in other ATP synthases[5–7], and in addition near the matrix side, 10 Arg66_c residues coordinate a ligand density, which is consistent with a pyrimidine ribonucleoside tri-phosphate (Fig. 1d). We assign this density as uridine-triphosphate (UTP), due to its large requirement in the mitochondrial RNA meta-bolism of African trypanosomes being a substrate for post-transcriptional RNA editing[25], and addition of poly-uridine tails to gRNAs and rRNAs[26,27], as well as due to low abundance of cytidine triphosphate (CTP)[28]. The nucleotide base is inserted between two Arg82_c residues, whereas the triphosphate region is coordinated by another five Arg82_c residues, with Tyr79_δ and Asn76_δ providing asymmetric coordination contacts. The presence of a nucleotide inside the *c*-ring is surprising, given the recent reports of phospho-lipids inside the *c*-rings in mammals[5,6] and ciliates[9], indicating that a range of different ligands can provide structural scaffolding.

## Peripheral stalk flexibility and distinct rotational states

The trypanosomal peripheral stalk displays a markedly different architecture compared to its yeast and mammalian counterparts. In the opisthokont complexes, the peripheral stalk is organized around the long *b*H2, which extends from the membrane ~15 nm into the matrix and attaches to OSCP at the top of F_1[5,7]. By contrast, *T. brucei* lacks the canonical *b*H2 and instead, helices 5-7 of divergent subunit-*d* and the C-terminal helix of extended subunit-*8* bind to a C-terminal extension of OSCP at the apical part of the peripheral stalk (Fig. 3a). The interaction between OSCP and subunit-*d* and -*8* is stabilized by soluble ATPTB3 and ATPTB4. The peripheral stalk is rooted to the membrane subcomplex by a transmembrane helix of subunit-*8*, wrapped on the matrix side by helices 8-11 of subunit-*d*. Apart from the canonical contacts at the top of F_1, the peripheral stalk is attached to the F_1 via an euglenozoa-specific C-terminal extension of OSCP, which contains a disordered linker and a terminal helix hairpin extending between the F_1-bound p18 and subunits -*d* and -*8* of the peripheral stalk (Fig. 3a and Supplementary Movies 2, 3). Another interaction of F_1 with the peripheral stalk occurs between the stacked C-terminal helices of subunit-*β* and -*d* (Fig. 3b), the latter of which structurally belongs to F_1 and is connected to the peripheral stalk via a flexible linker.

To assess whether the unusual peripheral stalk architecture influences the rotary mechanism, we analyzed 10 classes representing different rotational states. The three main states (1–3) result from three ~120° rotation steps of the rotor relatively to the static F_o. In all classes F_1 is in a similar conformation, corresponding to the catalytic dwell, observed previously also in the crystal structure of *T. brucei* F_1-ATPase[13]. In accordance with the ~120° rotation of the central stalk, the conformations and nucleotide occupancy of the catalytic interfaces of the individual *αβ* dimers differ between the main states, showing ADP and ATP in the "loose" and "tight" closed conformations, respectively, and empty nucleotide binding site in the "open" conformation. We identified five (1a–1e), four (2a–2d), and one (3) classes of the respective main states. The rotor positions of the rotational states 1a, 2a, and 3 are related by steps of 117°, 136°, and 107°, respectively. Throughout

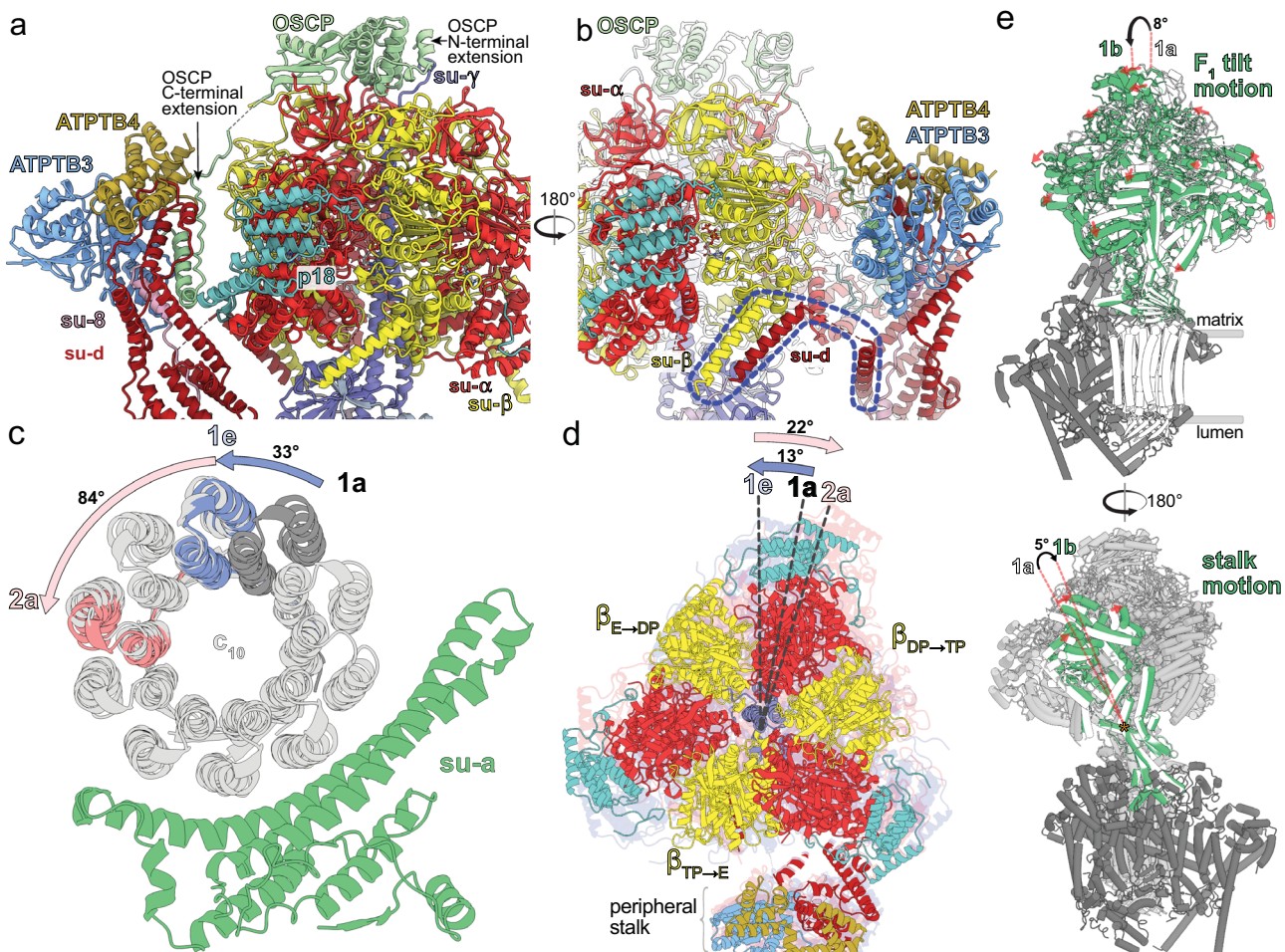

**Fig. 3 | A divergent peripheral stalk allows high flexibility during rotary catalysis. a** N-terminal OSCP extension provides a permanent central stalk attachment, while the C-terminal extension provides a phylum-specific attachment to the divergent peripheral stalk. **b** The C-terminal helices of subunits -$\beta$ and -$d$ provide a permanent F$_1$ attachment. **c** Substeps of the $c$-ring during transition from rotational state 1 to 2. **d** F$_1$ motion accommodating steps shown in (**c**). After advancing along with the rotor to state 1e, the F$_1$ rotates in the opposite direction when transitioning to state 2a. **e** Tilting motion of F$_1$ and accommodating bending of the peripheral stalk.

all the identified substeps of the rotational state 1 (classes 1a to 1e) the rotor turns by ~33°, which corresponds approximately to the advancement by one subunit-$c$ of the $c_{10}$-ring (Fig. 3c). While rotating along with the rotor, the F$_1$ headpiece lags behind, advancing by only ~13°. During the following transition from 1e to 2a, the rotor advances by ~84°, whereas the F$_1$ headpiece rotates ~22° in the opposite direction (Fig. 3d). This generates a counter-directional torque between the two motors, which is consistent with a power-stroke mechanism. This counter-directional torque may occur in all three main rotational state transitions. However, it was observed only in the main state 1, because it was captured in more substeps than the remaining two states, presumably as a consequence of the symmetry mismatch between the decameric $c$-ring and the $\alpha_3\beta_3$ hexamer[29]. Within the four classes of the state 2 the rotor advances by 23° and F$_1$ returns close to its position observed in class 1a, where it is found also in the only observed class of the state 3. Albeit with small differences in step size, this mechanism is consistent with a previous observation in the *Polytomella* ATP synthase[8]. However, due to its large, rigid peripheral stalk, the *Polytomella* ATP synthase mainly displays rotational substeps, whereas the *Trypanosoma* F$_1$ also displays a tilting motion of ~8° revealed by rotary states 1a and 1b (Fig. 3e and Supplementary Movie 2). The previously reported hinge motion between the N- and C-terminal domains of OSCP[8] is not found in our structures, instead, the conformational changes of the F$_1$/$c_{10}$-ring subcomplex are accommodated by a 5°

bending of the apical part of the peripheral stalk. (Fig. 3e and Supplementary Movies 2, 3). Together, the structural data indicate that the divergent peripheral stalk attachment confers greater conformational flexibility to the *T. brucei* ATP synthase.

## Lumenal proton half-channel is insulated by a lipid and contains ordered water molecules

The mechanism of proton translocation involves sequential protonation of E102 of subunits-$c$, rotation of the $c_{10}$-ring with neutralized E102$c$ exposed to the phospholipid bilayer, and release of protons on the other side of the membrane. The sites of proton binding and release are separated by the conserved R146 contributed by the horizontal helix H5 of subunit-$a$ and are accessible from the cristae lumen and mitochondrial matrix by aqueous half-channels (Fig. 4a). Together, R146 and the adjacent N209 coordinate a pair of water molecules in between helices H5 and H6 (Fig. 4b). A similar coordination has been observed in the *Polytomella* ATP synthase[8]. The coordination of water likely restricts the R146 to rotamers that extend towards the $c$-ring, with which it is thought to interact.

In our structure, the lumenal half-channel, which displays a local resolution of 2.55 Å (Supplementary Fig. 3), is filled with a network of resolved water densities, ending in a chain of five ordered water molecules (W1–W5; Fig. 4c–e). The presence of ordered water molecules in the aqueous channel is consistent with a Grotthuss-type

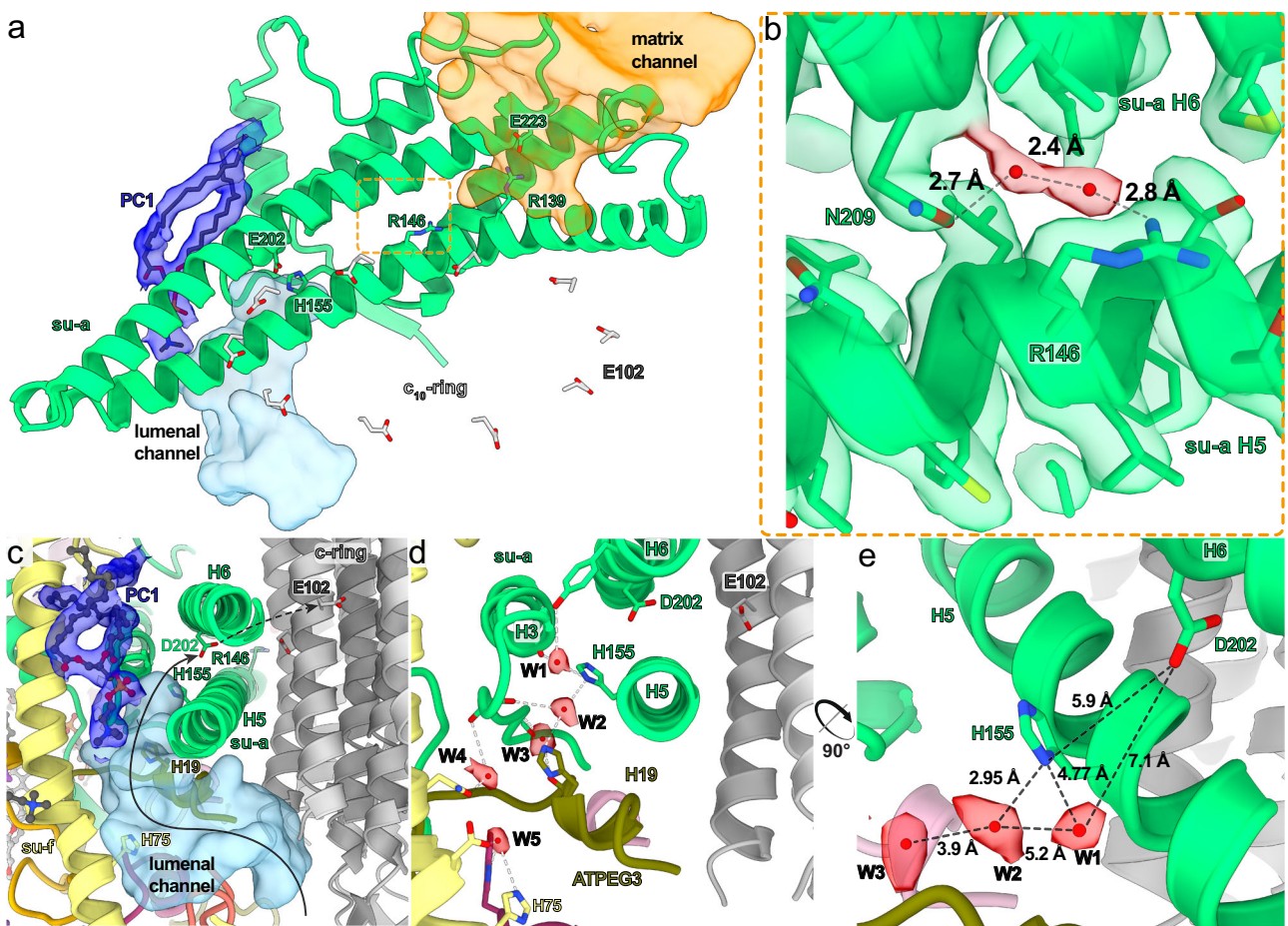

**Fig. 4 | The lumenal half-channel contains ordered water molecules and is confined by an $F_o$-bound lipid. a** Subunit-*a* (green) with the matrix (orange) and lumenal (light blue) channels, and an ordered phosphatidylcholine (PC1; blue). E102 of the $c_{10}$-ring shown in grey. **b** Close-up view of the highly conserved R146$_a$ and N209$_a$, which coordinate two water molecules between helices H5-6$_a$. **c** Sideview of the lumenal channel with proton pathway (light blue) and confining phosphatidylcholine (blue). **d** Chain of ordered water molecules in the lumenal channel. Distances between the W1–W5 (red) are 5.2, 3.9, 7.3, and 4.8 Å, respectively. **e** The ordered waters extend to H155$_a$, which likely mediates the transfer of protons to D202$_a$.

mechanism for proton transfer, which would not require long-distance diffusion of water molecules[5]. However, because some distances between the observed water molecules are too large for direct hydrogen bonding, proton transfer may involve both coordinated and disordered water molecules. The distance of 7 Å between the last resolved water (W1) and D202$_a$, the conserved residue that is thought to transfer protons to the *c*-ring, is too long for direct proton transfer. Instead, it may occur via the adjacent H155$_a$. Therefore, our structure resolves individual elements participating in proton transport (Fig. 4d, e).

The lumenal proton half-channel in the mammalian[5,6] and apicomplexan[11] ATP synthase is lined by the transmembrane part of *b*H2, which is absent in *T. brucei*. Instead, the position of *b*H2 is occupied by a fully ordered phosphatidylcholine in our structure (PC1; Fig. 4a, c). Therefore, a bound lipid replaces a proteinaceous element in the proton path.

**Subunit-*g* facilitates assembly of different ATP synthase oligomers**

Despite sharing a set of conserved $F_o$ subunits, the *T. brucei* ATP synthase dimer displays a markedly different dimer architecture compared to previously determined structures. First, its dimerization interface of 3600 Å² is smaller than that of the *E. gracilis* type-IV (10,000 Å²) and the *T. thermophila* type-III ATP synthases (16,000 Å²). Second, unlike mammalian and fungal ATP synthase, in which the

peripheral stalks extend in the plane defined by the two rotary axes, in our structure the monomers are rotated such that the peripheral stalks are offset laterally on the opposite sides of the plane. Due to the rotated monomers, this architecture is associated with a specific dimerization interface, where two subunit-*g* copies interact homotypically on the C$_2$ symmetry axis (Fig. 5a and Supplementary Movie 1). Both copies of H1-2$_g$ extend horizontally along the matrix side of the membrane, clamping against each other (Fig. 5c, e). This facilitates formation of contacts between an associated transmembrane helix of subunit-*e* with the neighbouring monomer via subunit-*a'* in the membrane, and -*f* in the lumen, thereby further contributing to the interface (Fig. 5b). Thus, the ATP synthase dimer is assembled via the subunit-*e/g* module. The C-terminal part of the subunit-*e* helix extends into the lumen, towards the ten-stranded β-barrel of the *c*-ring (Supplementary Fig. 7a). The terminal 23 residues are disordered with poorly resolved density connecting to the detergent plug of the *c*-ring β-barrel (Supplementary Fig. 7b). This resembles the lumenal C-terminus of subunit-*e* in the bovine structure[5], indicating a conserved interaction with the *c*-ring. In mammals, a mechanism, in which retraction of subunit-*e* upon calcium exposure pulls out the lipid plug and induces disassembly of the c-ring, which triggers permeability transition pore (PTP) opening, has been proposed[6].

The *e/g* module is held together by four bound cardiolipins in the matrix leaflet, anchoring it to the remaining $F_o$ region (Fig. 5c). The head groups of the lipids are coordinated by polar and charged

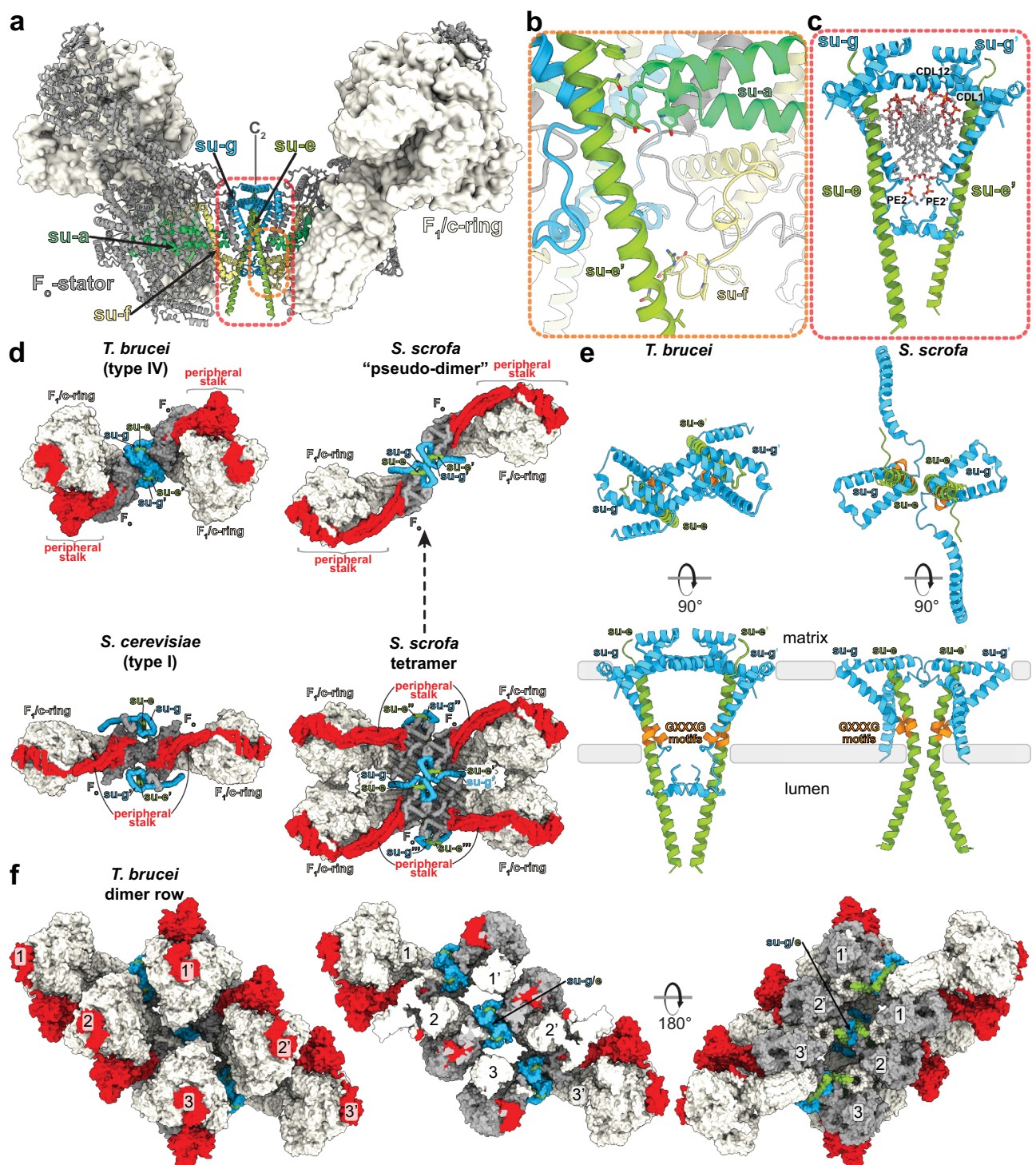

**Fig. 5 | The homotypic dimerization motif of subunit-*g* generates a conserved oligomerization module. a** Side view with dimerizing subunits coloured. The dimer interface is constituted by **b** subunit-*e'* contacting subunit-*a* in the membrane and subunit-*f* in the lumen, **c** subunits *e* and *g* from both monomers forming a subcomplex with bound lipids. **d** Subunit-*g* and -*e* form a dimerization motif in the trypanosomal (type-IV) ATP synthase dimer (this study), the same structural element forms the oligomerization motif in the porcine ATP synthase tetramer. The structural similarity of the pseudo-dimer (i.e., two diagonal monomers from adjacent dimers) in the porcine structure with the trypanosomal dimer suggests that type I and IV ATP synthase dimers have evolved through divergence from a common ancestor. **e** The dimeric subunit-*e/g* structures are conserved in *Sus scrofa* PDB 6ZNA (*S. scrofa* mitochondrial ATP synthase) and *T. brucei* (this work) and contain a conserved GXXXG motif (orange) mediating interaction of transmembrane helices. **f** Models of the ATP synthase dimers fitted into subtomogram averages of short oligomers[15]: matrix view, left; cut-through, middle, lumenal view, right; EMD-3560 (in situ structure of *T. brucei* mitochondrial ATP synthase).

residues with their acyl chains filling a central cavity in the membrane region at the dimer interface (Fig. 5c and Supplementary Fig. 5f). Cardiolipin binding has previously been reported to be obligatory for dimerization in secondary transporters[30] and the depletion of

cardiolipin synthase resulted in reduced levels of ATP synthase in bloodstream trypanosomes[14].

Interestingly, for yeasts, early blue native gel electrophoresis[31] and subtomogram averaging studies[2] suggested subunit-*g* as potentially

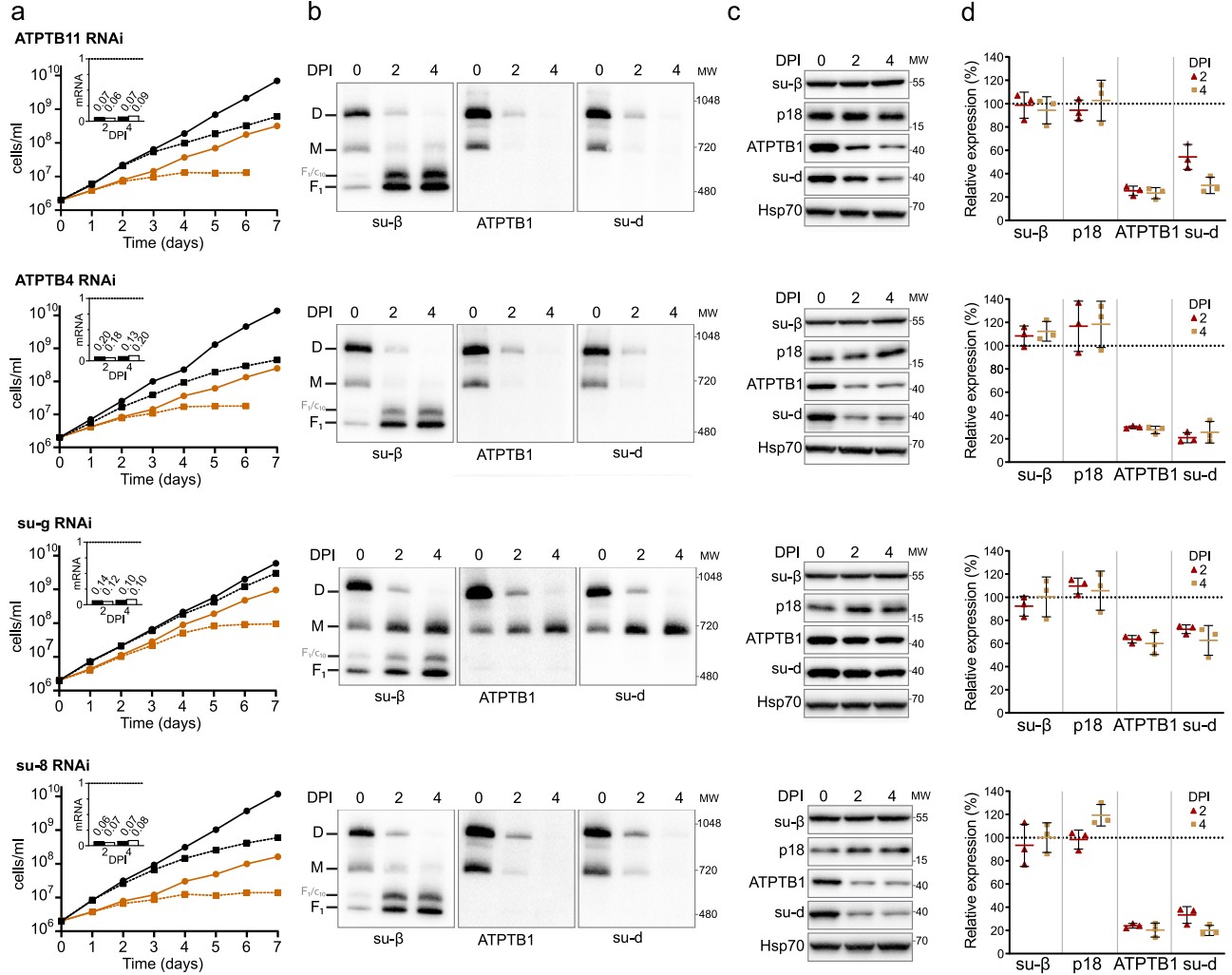

**Fig. 6 | RNAi knockdown of subunit-*g* results in monomerization of ATP synthase. a** Growth curves of non-induced (solid lines) and tetracycline-induced (dashed lines) RNAi cell lines grown in the presence (black) or absence (brown) of glucose. The insets show relative levels of the respective target mRNA at indicated days post-induction (DPI) normalized to the levels of 18S rRNA (black bars) or β-tubulin (white bars). **b** Immunoblots of mitochondrial lysates from indicated RNAi cell lines resolved by BN-PAGE probed with antibodies against indicated ATP synthase subunits (*n* = 2). Positions of molecular weight (MW) marker are shown. **c** Immunoblots of whole cell lysates from indicated RNAi cell lines probed with indicated antibodies (*n* = 3). Positions of MW marker are shown. **d** Quantification of three independent replicates of immunoblots in (**c**). Values were normalized to the signal of the loading control Hsp70 and to non-induced cells. Plots show individual values, means, and standard deviations (SD; error bars).

dimer-mediating, however the *e/g* modules are located laterally opposed on either side of the dimer long axis, in the periphery of the complex, ~8.5 nm apart from each other. Because the *e/g* modules do not interact directly within the yeast ATP synthase dimer, they have been proposed to serve as membrane-bending elements, whereas the major dimer contacts are formed by subunit-*a* and -*i/j*[7]. In mammals, the *e/g* module occupies the same position as in yeasts, forming the interaction between two diagonal monomers in a tetramer[5,6,32], as well as between parallel dimers[33]. The comparison with our structure shows that the overall organization of the intra-dimeric trypanosomal and inter-dimeric mammalian *e/g* module is structurally similar (Fig. 5d). Furthermore, kinetoplastid parasites and mammals share conserved GXXXG motifs in subunit-*e*[34] and -*g* (Supplementary Fig. 8), which allow close interaction of their transmembrane helices (Fig. 5e), providing further evidence for subunit homology. However, while the mammalian ATP synthase dimers are arranged perpendicularly to the long axis of their rows along the edge of cristae[35], the *T. brucei* dimers on the rims of discoidal cristae are inclined ~45° to the row axis[15]. Therefore, the *e/g* module occupies equivalent positions in the rows of both evolutionary distant groups (Fig. 5f and ref. 33).

## Subunit-*g* retains the dimer but is not essential for the catalytic monomer

To validate structural insights, we knocked down each individual $F_o$ subunit by inducible RNA interference (RNAi). All target mRNAs dropped to 5–20% of their original levels after two and four days of induction (Fig. 6a and Supplementary Fig. 9a). Western blot analysis of whole-cell lysates resolved by denaturing electrophoresis revealed decreased levels of $F_o$ subunits ATPB1 and -*d* suggesting that the integrity of the $F_o$ moiety depends on the presence of other $F_o$ subunits (Fig. 6c, d). Immunoblotting of mitochondrial complexes resolved by blue native polyacrylamide gel electrophoresis (BN-PAGE) with antibodies against $F_1$ and $F_o$ subunits revealed a strong decrease or nearly complete loss of dimeric and monomeric forms of ATP synthases four days after induction of RNAi of most subunits (*b*, *e*, *f*, *i/j*, *k*, 8, ATPTB3, ATPTB4, ATPTB6, ATPTB11, ATPTB12, ATPEG3, and ATPEG4), documenting an increased instability of the enzyme or defects in its assembly. Simultaneous accumulation in $F_1$-ATPase, as observed by BN-PAGE, demonstrated that the catalytic moiety remains intact after the disruption of the peripheral stalk or the membrane subcomplex (Fig. 6b–d and Supplementary Fig. 9b).

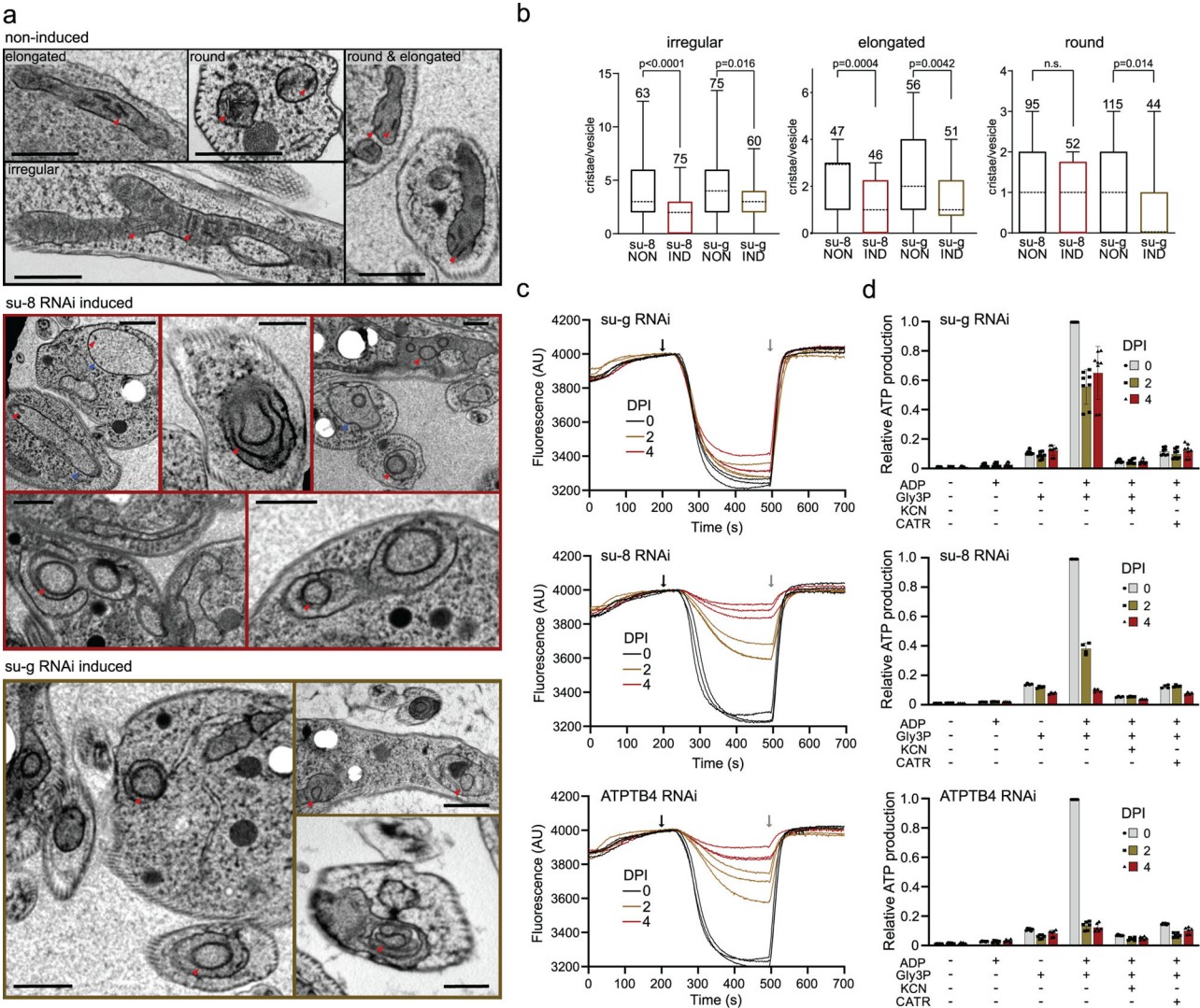

**Fig. 7 | Monomerization of ATP synthase by subunit-*g* knockdown results in aberrant mitochondrial ultrastructure but does not abolish catalytic activity. a** Transmission electron micrographs of sections of non-induced or 4 days induced RNAi cell lines. At least 70 micrographs were obtained in each category. Mitochondrial membranes and cristae are marked with blue and red arrowheads, respectively. Top panel shows examples of irregular, elongated and round cross-sections of mitochondria quantified in (**b**). Scale bars: 500 nm. **b** Cristae numbers per vesicle from indicated induced (IND) or non-induced (NON) cell lines counted separately in irregular, elongated and round mitochondrial cross-section. Boxes and whiskers show 25th to 75th and 5th to 95th percentiles, respectively. The

numbers of analyzed cross-sections are indicated for each data point. Unpaired two-sided t-test, *p*-values are shown in the graph. **c** Mitochondrial membrane polarization capacity of non-induced or RNAi-induced cell lines two and four DPI measured by Safranine O. Black and grey arrow indicate addition of ATP and oligomycin, respectively. **d** ATP production in permeabilized non-induced (0) or RNAi-induced cells 2 and 4 DPI in the presence of indicated substrates and inhibitors. The graphs show individual values of two technical replicates of $n = 2$ (subunit-*8*), $n = 3$ (ATPTB4), or $n = 4$ (subunit-*g*) independent experiments and means (bars) and SD (error bars) of the averaged values of the technical replicates. Gly3P DL-glycerol phosphate; KCN potassium cyanide; CATR carboxyatractyloside.

In contrast to the other targeted $F_o$ subunits, the downregulation of subunit-*g* with RNAi resulted in a specific loss of dimeric complexes with concomitant accumulation of monomers (Fig. 6b), indicating that it is required for dimerization, but not for the assembly and stability of the monomeric $F_1F_o$ ATP synthase units. Transmission electron microscopy of thin cell sections revealed that the ATP synthase monomerization in the subunit-*g*[RNAi] cell line had the same effect on mitochondrial ultrastructure as nearly complete loss of monomers and dimers upon knockdown of subunit-*8*. Both cell lines exhibited decreased cristae counts and aberrant cristae morphology (Fig. 7a, b), including the appearance of round shapes reminiscent of structures detected upon deletion of subunit-*g* or -*e* in *Saccharomyces cerevisiae*[1]. These results indicate that monomerization prevents the trypanosomal ATP synthase from assembling into short helical rows on the rims of the discoidal cristae[15], as has been reported for impaired oligomerization in counterparts from other eukaryotes[2,36].

Despite the altered mitochondrial ultrastructure, the subunit-*g*[RNAi] cells showed only a very mild growth phenotype, in contrast to all other RNAi cell lines that exhibited steadily slowed growth from day three to four after the RNAi induction (Fig. 7a and Supplementary Fig. 9a). This is consistent with the growth defects observed after the ablation of $F_o$ subunit ATPTB1[19] and $F_1$ subunits-α and p18[12]. Thus, the monomerization of ATP synthase upon subunit-*g* ablation had only a negligible effect on the fitness of trypanosomes cultured in glucose-rich medium, in which ATP production by substrate level phosphorylation partially compensates for compromised oxidative phosphorylation[37].

Measurement of oligomycin-sensitive ATP-dependent mitochondrial membrane polarization by safranin O assay in permeabilized cells showed that the proton pumping activity of the ATP synthase in the induced subunit-*g*[RNAi] cells is negligibly affected, demonstrating that the monomerized enzyme is catalytically functional. By contrast, RNAi

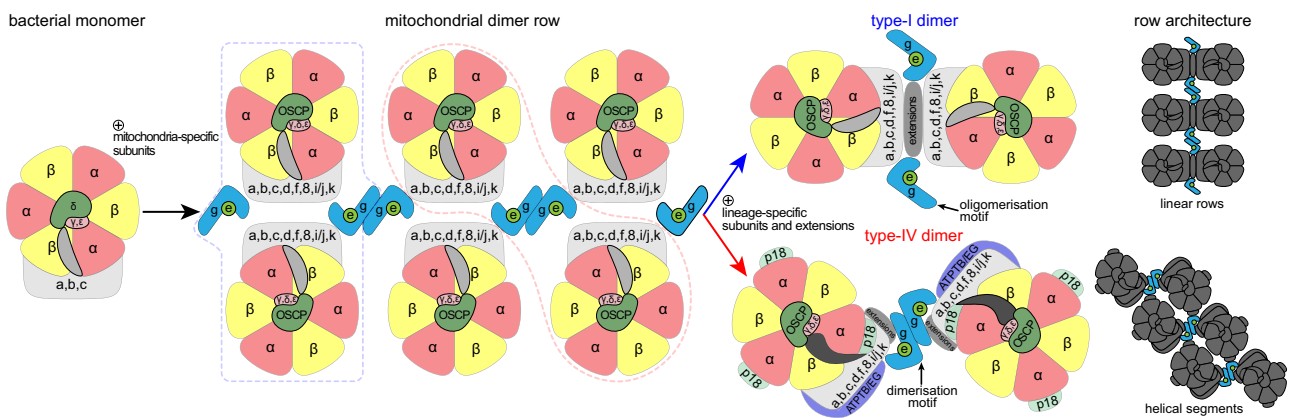

**Fig. 8 | The subunit-*e/g* module is an ancestral oligomerization motif of ATP synthase.** Schematic model of the evolution of type-I and IV ATP synthases. Mitochondrial ATP synthases are derived from a monomeric complex of proteobacterial origin. In a mitochondrial ancestor, acquisition of mitochondria-specific subunits, including the subunit-*e/g* module resulted in the assembly of ATP synthase double rows, the structural basis for cristae biogenesis. Through divergence, different ATP synthase architectures evolved, with the subunit-*e/g* module functioning as an oligomerization (type I) or dimerization (type IV) motif, resulting in distinct row assemblies between mitochondrial lineages.

downregulation of subunit-*8*, ATPTB4 and ATPTB11, and ATPTB1 resulted in a strong decline of the mitochondrial membrane polarization capacity, consistent with the loss of both monomeric and dimeric ATP synthase forms (Fig. 7c). Accordingly, knockdown of the same subunits resulted in inability to produce ATP by oxidative phosphorylation (Fig. 7d). However, upon subunit-*g* ablation the ATP production was affected only partially, confirming that the monomerized ATP synthase remains catalytically active. The ~50% drop in ATP production of subunit-*g*^RNAi cells can be attributed to the decreased oxidative phosphorylation efficiency due to the impaired cristae morphology. Indeed, when cells were cultured in the absence of glucose, enforcing the need for oxidative phosphorylation, knockdown of subunit-*g* results in a growth arrest, albeit one to two days later than knockdown of all other tested subunits (Fig. 6a). The data show that dimerization is critical when oxidative phosphorylation is the predominant source of ATP.

## Discussion

Our structure of the mitochondrial ATP synthase dimer from the mammalian parasite *T. brucei* offers new insight into the mechanism of membrane shaping, rotary catalysis, and proton transfer. Considering that trypanosomes belong to an evolutionarily divergent group of Kinetoplastida, the ATP synthase dimer has several interesting features that differ from other dimer structures. The subunit-*b* found in bacterial and other mitochondrial F-type ATP synthases appears to be highly reduced to a single transmembrane helix *b*H1. The long *b*H2, which constitutes the central part of the peripheral stalk in other organisms, and is also involved in the composition of the lumenal proton half-channel, is completely absent in *T. brucei*. Interestingly, the position of *b*H2 in the proton half channel is occupied by a fully ordered phosphatidylcholine molecule that replaces a well-conserved proteinaceous element in the proton path. However, this replacement is not a common trait of all type-IV ATP synthases, because subunit-*b* in *E. gracilis* contains the canonical bH2 but lacks bH1[10]. Thus, while subunit-*b* is conserved in Euglenozoa, the lineages of *T. brucei* and *E. gracilis* retained its different non-overlapping structural elements (Fig. 2c). Lack of bH2 in *T. brucei* also affects composition of the peripheral stalk in which the divergent subunit-*d* and subunit-*8* binds directly to a C-terminal extension of OSCP, indicating a remodeled peripheral stalk architecture. The peripheral stalk contacts the F₁ headpiece at several positions conferring greater conformational flexibility to the ATP synthase.

Using the structural and functional data, we also identified a conserved structural element of the ATP synthase that is responsible for its multimerization. Particularly, subunit-*g* is required for the dimerization, but dispensable for the assembly of the F₁Fₒ monomers. Although the monomerized enzyme is catalytically competent, the inability to form dimers results in defective cristae structure, and consequently leads to compromised oxidative phosphorylation and cease of proliferation. The cristae-shaping properties of mitochondrial ATP synthase dimers are critical for sufficient ATP production by oxidative phosphorylation, but not for other mitochondrial functions, as demonstrated by the lack of growth phenotype of subunit-*g*^RNAi cells in the presence of glucose. Thus, trypanosomal subunit-*g* depletion strain represents an experimental tool to assess the roles of the enzyme's primary catalytic function and mitochondria-specific membrane-shaping activity, highlighting the importance of the latter for oxidative phosphorylation.

Based on our data and previously published structures, we propose an ancestral state with double rows of ATP synthase monomers connected by *e/g* modules longitudinally and by other Fₒ subunits transversally. During the course of evolution, different pairs of adjacent ATP synthase monomer units formed stable dimers in individual lineages (Fig. 8). This gave rise to the highly divergent type-I and type-IV ATP synthase dimers with subunit-*e/g* modules serving either as oligomerization or dimerization motives, respectively. Because trypanosomes belong to the deep-branching eukaryotic supergroup Discoba, the proposed arrangement might have been present in the last eukaryotic common ancestor. Although sequence similarity of subunit-*g* is low and restricted to the single transmembrane helix, we found homologues of subunit-*g* in addition to Opisthokonta and Discoba also in Archaeplastida and Amoebozoa, which represent other eukaryotic supergroups, thus supporting the ancestral role in oligomerization (Supplementary Fig. 8). Taken together, our analysis reveals that mitochondrial ATP synthases that display markedly diverged architecture share the ancestral structural module that promotes oligomerization.

## Methods

### Cell culture and isolation of mitochondria

*T. brucei* procyclic strains were cultured in SDM-79 medium supplemented with 10% (v/v) fetal bovine serum. For growth curves in glucose-free conditions, cells were grown in SDM-80 medium with 10% dialyzed FBS. RNAi cell lines were grown in presence of 2.5 µg/ml phleomycin and 1 µg/ml puromycin. For ATP synthase purification, mitochondria were isolated from the Lister strain 427. Typically, 1.5 × 10¹¹ cells were harvested, washed in 20 mM sodium phosphate buffer pH 7.9 with 150 mM NaCl and 20 mM glucose, resuspended in

hypotonic buffer 1 mM Tris-HCl pH 8.0, 1 mM EDTA, and disrupted by 10 strokes in a 40 ml Dounce homogenizer. The lysis was stopped by immediate addition of sucrose to 0.25 M. Crude mitochondria were pelleted (15 min at $16,000 \times g$, 4 °C), resuspended in 20 mM Tris-HCl pH 8.0, 250 mM sucrose, 5 mM MgCl₂, 0.3 mM CaCl₂ and treated with 5 µg/ml DNase I. After 60 min on ice, one volume of the STE buffer (20 mM Tris-HCl pH 8.0, 250 mM sucrose, 2 mM EDTA) was added and mitochondria were pelleted (15 min at $16000 \times g$, 4 °C). The pellet was resuspended in 60% (v/v) Percoll in STE and loaded on six linear 10–35% Percoll gradients in STE in polycarbonate tubes for SW28 rotor (Beckman). Gradients were centrifuged for 1 h at $104000 \times g$, 4 °C. The middle phase containing mitochondrial vesicles (15–20 ml per tube) was collected, washed four times in the STE buffer, and pellets were snap-frozen in liquid nitrogen and stored at −80 °C.

## Plasmid construction and generation of RNAi cell lines

To downregulate ATP synthase subunits by RNAi, DNA fragments corresponding to individual target sequences were amplified by PCR from Lister 427 strain genomic DNA using forward and reverse primers extended with restriction sites *Xho*I&*Kpn*I and *Xba*I&*Bam*HI, respectively (Supplementary Table 3). Each fragment was inserted into the multiple cloning sites 1 and 2 of pAZ0055 vector, derived from pRP^HYG-iSL (courtesy of Sam Alsford) by replacement of hygromycine resistance gene with phleomycine resistance gene, with restriction enzymes *Kpn*I/*Bam*HI and *Xho*I/*Xba*I, respectively. Resulting constructs with tetracycline inducible T7 polymerase driven RNAi cassettes were linearized with *Not*I and transfected into a cell line derived from the Lister strain 427 by integration of the SmOx construct for expression of T7 polymerase and the tetracycline repressor[38] into the β-tubulin locus. RNAi was induced in selected semi-clonal populations by addition of 1 µg/ml tetracycline and the downregulation of target mRNAs was verified by quantitative RT-PCR 2 and 4 days post induction. The total RNA isolated by an RNeasy Mini Kit (Qiagen) was treated with 2 µg of DNase I, and then reverse transcribed to cDNA with TaqMan Reverse Transcription kit (Applied Biosciences). qPCR reactions were set with Light Cycler 480 SYBR Green I Master mix (Roche), 2 µl of cDNA, and 0.3 µM primers (Supplementary Table 3) and run on LightCycler 480 (Roche). Relative expression of target genes was calculated using - ΔΔCt method with 18S rRNA or β-tubulin as endogenous reference genes and normalized to noninduced cells.

## Denaturing and blue native polyacrylamide electrophoresis and immunoblotting

Whole cell lysates for denaturing sodium dodecyl sulphate polyacrylamide electrophoresis (SDS-PAGE) were prepared from cells resuspended in PBS buffer (10 mM phosphate buffer, 130 mM NaCl, pH 7.3) by addition of 3× Laemmli buffer (150 mM Tris pH 6.8, 300 mM 1,4-dithiothreitol, 6% (w/v) SDS, 30% (w/v) glycerol, 0.02% (w/v) bromophenol blue) to final concentration of $1 \times 10^7$ cells in 30 µl. The lysates were boiled at 97 °C for 10 min and stored at −20 °C. For immunoblotting, lysates from $3 \times 10^6$ cells were separated on 4–20% gradient Tris-glycine polyacrylamide gels (BioRad 4568094), electroblotted onto a PVDF membrane (Pierce 88518), and probed with respective antibodies (Supplementary Table 4). Membranes were incubated with the Clarity Western ECL substrate (BioRad 1705060EM) and chemiluminescence was detected on a ChemiDoc instrument (BioRad). Band intensities were quantified densitometrically using the ImageLab software. The levels of individual subunits were normalized to the signal of mtHsp70.

Blue native PAGE (BN-PAGE) was performed as described earlier[12] with following modifications. Crude mitochondrial vesicles from $2.5 \times 10^8$ cells were resuspended in 40 µl of Solubilization buffer A (2 mM ε-aminocaproic acid (ACA), 1 mM EDTA, 50 mM NaCl, 50 mM Bis-Tris/HCl, pH 7.0) and solubilized with 2% (w/v) dodecylmaltoside (β-DDM)

for 1 h on ice. Lysates were cleared at $16,000 \times g$ for 30 min at 4 °C and their protein concentration was estimated using bicinchoninic acid assay. For each time point, a volume of mitochondrial lysate corresponding to 4 µg of total protein was mixed with 1.5 µl of loading dye (500 mM ACA, 5% (w/v) Coomassie Brilliant Blue G-250) and 5% (w/v) glycerol and with 1 M ACA until a final volume of 20 µl/well and resolved on a native PAGE 3–12% Bis-Tris gel (Invitrogen). After the electrophoresis (3 h, 140 V, 4 °C), proteins were transferred by electroblotting onto a PVDF membrane (2 h, 100 V, 4 °C, stirring), followed by immunodetection with an appropriate antibody (Supplementary Table 4).

## Mitochondrial membrane polarization measurement

The capacity to polarize mitochondrial membrane was determined fluorometrically employing safranin O dye (Sigma S2255) in permeabilized cells. For each sample, $2 \times 10^7$ cells were harvested and washed with ANT buffer (8 mM KCl, 110 mM K-gluconate, 10 mM NaCl, 10 mM free-acid Hepes, 10 mM K₂HPO₄, 0.015 mM EGTA potassium salt, 10 mM mannitol, 0.5 mg/ml fatty acid-free BSA, 1.5 mM MgCl₂, pH 7.25). The cells were permeabilized by 8 µM digitonin in 2 ml of ANT buffer containing 5 µM safranin O. Fluorescence was recorded for 700 s in a Hitachi F-7100 spectrofluorimeter (Hitachi High Technologies) at a 5 Hz acquisition rate, using 495 and 585 nm excitation and emission wavelengths, respectively. 1 mM ATP (PanReac AppliChem A1348,0025) and 10 µg/ml oligomycin (Sigma O4876) were added after 230 s and 500 s, respectively. Final addition of the uncoupler SF 6847 (250 nM; Enzo Life Sciences BML-EI215-0050) served as a control for maximal depolarization. All experiments were performed at room temperature and constant stirring.

## ATP production assay

ATP production in digitonin-isolated mitochondria was performed as described previously[39]. Briefly, $1 \times 10^8$ cells per time point were lysed in SoTE buffer (600 mM sorbitol, 2 mM EDTA, 20 mM Tris-HCl, pH 7.75) containing 0.015% (w/v) digitonin for 5 min on ice. After centrifugation (3 min, $4000 \times g$, 4 °C), the soluble cytosolic fraction was discarded and the organellar pellet was resuspended in 75 µl of ATP production assay buffer (600 mM sorbitol, 10 mM MgSO₄, 15 mM potassium phosphate buffer pH 7.4, 20 mM Tris-HCl pH 7.4, 2.5 mg/ml fatty acid-free BSA). ATP production was induced by addition of 20 mM DL-glycerol phosphate (sodium salt) and 67 µM ADP. Control samples were preincubated with the inhibitors potassium cyanide (1 mM) and carboxyatractyloside (6.5 µM) for 10 min at room temperature. After 30 min at room temperature, the reaction was stopped by the addition of 1.5 µl of 70% perchloric acid. The concentration of ATP was estimated using the Roche ATP Bioluminescence Assay Kit HS II in a Tecan Spark plate reader. The luminescence values of the RNAi induced samples were normalized to that of the corresponding noninduced sample.

## Thin sectioning and transmission electron microscopy

The samples were centrifuged and pellet was transferred to the specimen carriers which were completed with 20% BSA and immediately frozen using high pressure freezer Leica EM ICE (Leica Microsystems). Freeze substitution was performed in the presence of 2% osmium tetroxide diluted in 100% acetone at −90 °C. After 96 h, specimens were warmed to −20 °C at a slope 5 °C/h. After the next 24 h, the temperature was increased to 3 °C (3 °C/h). At room temperature, samples were washed in acetone and infiltrated with 25, 50, 75% acetone/resin EMbed 812 (EMS) mixture 1 h at each step. Finally, samples were infiltrated in 100% resin and polymerized at 60 °C for 48 h. Ultrathin sections (70 nm) were cut using a diamond knife, placed on copper grids, and stained with uranyl acetate and lead citrate. TEM micrographs were taken with Mega View III camera (SIS) using a JEOL 1010 TEM operating at an accelerating voltage of 80 kV.

## Purification of *T. brucei* ATP synthase dimers

Mitochondria from $3 \times 10^{11}$ cells were lysed by 1% (w/v) β-DDM in 60 ml of 20 mM Bis-tris propane pH 8.0 with 10% glycerol and EDTA-free Complete protease inhibitors (Roche) for 20 min at 4 °C. The lysate was cleared by centrifugation at $30,000 \times g$ for 20 min at 4 °C and adjusted to pH 6.8 by drop-wise addition of 1 M 3-(N-morpholino) propanesulfonic acid pH 5.9. Recombinant TbIF$_1$ without dimerization region, whose affinity to F$_1$-ATPase was increased by N-terminal truncation and substitution of tyrosine 36 with tryptophan[20], with a C-terminal glutathione S-transferase (GST) tag (TbIF$_1$(9-64)-Y36W-GST) was added in approximately 10-fold molar excess over the estimated content of ATP synthase. Binding of TbIF$_1$ was facilitated by the addition of neutralized 2 mM ATP with 4 mM magnesium sulphate. After 5 min, sodium chloride was added to 100 mM, the lysate was filtered through a 0.2 µm syringe filter and immediately loaded on 5 ml GSTrap HP column (Cytiva) equilibrated in 20 mM Bis-Tris-Propane pH 6.8 binding buffer containing 0.1% (w/v) glyco-diosgenin (GDN; Avanti Polar Lipids), 10% (v/v) glycerol, 100 mM sodium chloride, 1 mM tris(2-carboxyethyl)phosphine (TCEP), 1 mM ATP, 2 mM magnesium sulphate, 15 µg/ml cardiolipin, 50 µg/ml 1-palmitoyl-2-oleoyl-sn-glycero-3-phosphocholine (POPC), 25 µg/ml 1-palmitoyl-2-oleoyl-sn-glycero-3-phosphoethanolamine (POPE) and 10 µg/ml 1-palmitoyl-2-oleoyl-sn-glycero-3-[phospho-rac-(1-glycerol)] (POPG). All phospholipids were purchased from Avanti Polar Lipids (catalogue numbers 840012C, 850457C, 850757C, and 840757, respectively). ATP synthase was eluted with a gradient of 20 mM reduced glutathione in Tris pH 8.0 buffer containing the same components as the binding buffer. Fractions containing ATP synthase were pooled and concentrated to 150 µl on Vivaspin centrifugal concentrator with 30 kDa molecular weight cut-off. The sample was fractionated by size exclusion chromatography on a Superose 6 Increase 3.2/300 GL column (Cytiva) equilibrated in a buffer containing 20 mM Tris pH 8.0, 100 mM sodium chloride, 2 mM magnesium chloride, 0.1% (w/v) GDN, 3.75 µg/ml cardiolipin, 12.5 µg/ml POPC, 6.25 µg/ml POPE and 2.5 µg/ml POPG at 0.03 ml/min. Fractions corresponding to ATP synthase were pooled, supplemented with 0.05% (w/v) β-DDM that we and others experimentally found to better preserve dimer assemblies in cryo-EM[40], and concentrated to 50 µl.

## Preparation of cryo-EM grids and data collection

Samples were vitrified on glow-discharged Quantifoil R1.2/1.3 Au 300-mesh grids after blotting for 3 s, followed by plunging into liquid ethane using a Vitrobot Mark IV. 5199 movies were collected using EPU 1.9 on a Titan Krios (ThermoFisher Scientific) operated at 300 kV at a nominal magnification of 165 kx (0.83 Å/pixel) with a Quantum K2 camera (Gatan) using a slit width of 20 eV. Data was collected with an exposure rate of 3.6 electrons/px/s, a total exposure of 33 electrons/Å$^2$, and 20 frames per movie.

## Image processing

Image processing was performed within the Scipion 2 framework[41], using RELION-3.0 unless specified otherwise. Movies were motion-corrected using the RELION implementation of the MotionCor2. 294,054 particles were initially picked using reference-based picking in Gautomatch (http://www.mrc-lmb.cam.ac.uk/kzhang/Gautomatch) and Contrast-transfer function parameters were using GCTF[42]. Subsequent image processing was performed in RELION-3.0 and 2D and 3D classification was used to select 100,605 particles, which were then extracted in an unbinned 560-pixel box (Fig. S1). An initial model of the ATP synthase dimer was obtained using de novo 3D model generation. Using masked refinement with applied C$_2$ symmetry, a 2.7 Å structure of the membrane region was obtained following per-particle CTF refinement and Bayesian polishing. Following C$_2$-symmetry expansion and signal subtraction of one monomer, a 3.7 Å map of the peripheral stalk was obtained. Using 3D classification ($T = 100$) of aligned particles, with a

mask on the F$_1$/$c$-ring region, 10 different rotational substates were then separated, and maps at 3.5–4.3 Å resolution were obtained using 3D refinement. The authors note that the number of classes identified in this study likely reflects the limited number of particles, rather than the complete conformational space of the complex. By combining particles from all states belonging to main rotational state 1, a 3.7 Å map of the rotor and a 3.2 Å consensus map of the complete ATP synthase dimer with both rotors in main rotational state 1 were obtained.

## Model building, refinement, and data visualization

An initial atomic model of the static F$_o$ membrane region was built automatically using Bucaneer[43]. Subunits were subsequently assigned directly from the cryo-EM map, 15 of them corresponding to previously identified *T. brucei* ATP synthase subunits[21], while three subunits (ATPTB14, ATPEG3, and ATPEG4) were identified herein using BLAST searches. Manual model building was performed in Coot 0.9.5[44] using the *T. brucei* F$_1$ PDB 6F5D [https://www.rcsb.org/structure/6F5D] (*T. brucei* F$_1$)[13] and homology models[45] of the *E. gracilis* OSCP and $c$-ring PDB 6TDU [https://www.rcsb.org/structure/6TDU] (*E. gracilis* mitochondrial ATP synthase)[10] as starting models. Ligands were manually fitted to the map and restraints were generated by the GRADE server (http://grade.globalphasing.org). Cardiolipins were assigned based on the presence of a characteristic elongated density branched on both termini, corresponding to two phosphatidyl groups linked by the central glycerol bridge. Monophosphatidyl lipids were assigned based on their headgroup densities. Characteristic tetrahedral shapes of densities of choline groups served to distinguish phosphatidylcholines from elongated phosphatidylethanolamine head groups (Supplementary Fig. 5g, h). Real-space refinement was performed in PHENIX 1.17.1 using auto-sharpened, local-resolution-filtered maps of the membrane region, peripheral stalk tip, $c$-ring/central stalk, and F$_1$F$_o$ monomers in different rotational states, respectively, using secondary structure restraints. Model statistics were generated using MolProbity[46] and EMRinger[47] Finally, the respective refined models were combined into a composite ATP synthase dimer model and real-space refined against the local-resolution-filtered consensus ATP synthase dimer map with both monomers in rotational state 1, applying reference restraints. Figures of the structures were prepared using ChimeraX 0.91[48], the proton half-channels were traced using HOLLOW[49].

## Reporting summary

Further information on research design is available in the Nature Research Reporting Summary linked to this article.

# Data availability

The atomic coordinates generated in this study have been deposited in the Protein Data Bank (PDB) under the accession codes: 8AP7 (membrane-region), 8AP8 (peripheral stalk), 8AP9 (rotor), 8AP6 (F1Fo dimer), 8APA (rotational state 1a), 8APB (rotational state 1b), 8APC (rotational state 1c), 8APD (rotational state 1d), 8APE (rotational state 1e), 8APF (rotational state 2a), 8APG (rotational state 2b), 8APH (rotational state 2c), 8APJ (rotational state 2d), 8APK (rotational state 3). The local resolution filtered cryo-EM maps, half maps, masks and FSC-curves have been deposited in the Electron Microscopy Data Bank under accession codes: EMD-15560 (membrane-region), EMD-15561 (peripheral stalk), EMD-15562 (rotor), EMD-15559 (F$_1$F$_o$ dimer), EMD-15563 (rotational state 1a), EMD-15564 (rotational state 1b), EMD-15565 (rotational state 1c), EMD-15566 (rotational state 1d), EMD-15567 (rotational state 1e), EMD-15568 (rotational state 2a), EMD-15570 (rotational state 2b), EMD-15571 (rotational state 2c), EMD-15572 (rotational state 2d), EMD-15573 (rotational state 3). The TEM micrographs of thin cell sections are available from the authors upon request. All other data are available in the article, Supplementary Information or the Source Data file. Source data are provided with this paper.

The atomic coordinates that were used in this study: 6TDU (*E. gracilis* mitochondrial ATP synthase), 6TDV (*E. gracilis* mitochondrial ATP synthase, membrane region), 6B2Z (*S. cerevisiae* mitochondrial ATP synthase), 6F5D (*T. brucei* F$_1$), 6ZNA (*S. scrofa* mitochondrial ATP synthase) Source data are provided with this paper.

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

## Acknowledgements

We are grateful to John E. Walker and Martin G. Montgomery for invaluable assistance with ATP synthase purification in the initial stage of the project. We acknowledge cryo-electron microscopy and tomography core facility of CIISB, Instruct-CZ Centre, supported by MEYS CR (LM2018127). This work was supported by the Czech Science Foundation grants number 18-17529S to A.Z. and 20-04150Y to O.G. and by European Regional Development Fund (ERDF) and Ministry of Education, Youth and Sport (MEYS) project CZ.02.1.01/0.0/0.0/16_019/0000759 to A.Z., Swedish Foundation for Strategic Research (FFL15:0325), Ragnar Söderberg Foundation (M44/16), European Research Council (ERC-2018-StG-805230), Knut and Alice Wallenberg Foundation (2018.0080), and EMBO Young Investigator Programme to A.A.

## Author contributions

A.Z. and A.A. conceived and designed the work. O.G. prepared the sample for cryo-EM. O.G. and A.M. performed initial screening. A.M. processed the cryo-EM data and built the model. O.G., A.M., and A.A. analyzed the structure. B.P., C.H.Y., M.J., M.S., O.G., D.H., and A.Z. performed biochemical analysis. O.G., A.M., A.A., and A.Z. interpreted the data. O.G., A.M., A.A., and A.Z. wrote and revised the manuscript. All authors contributed to the analysis and approved the final version of the manuscript.

## Funding

## Competing interests

The authors declare no competing interests.
