## [Peer Review File · Nature Communications]

An ancestral interaction module promotes oligomerization in divergent mitochondrial ATP synthasesREVIEWER COMMENTS

Reviewer #1 (Remarks to the Author):

To generate mitochondrial inner membrane curvature and maintain high level of ATP production, the mitochondrial ATP synthase forms dimers which further arrange into oligomers. In this work, Gahura et al. determined the cryo-EM structure of the ATP synthase dimer from *Trypanosoma brucei*, identified both protein and lipid components, revealed a set of rotational states, and proposed subunit-e/g module as an ancestral oligomerization motif of ATP synthases. The functional importance of subunit-g was supported by their RNAi knockdown, mitochondrial ultrastructure, membrane polarization measurement, and ATP production assay. This manuscript is well written with clearly presented figures, and most results are properly interpreted. The results are interesting to the community of ATP synthase research as well as broader membrane protein biology.

I have a few comments and questions listed below.

For the assignment of cardiolipin molecules, particularly those in Fig. 1c, the author should discuss the possibility of other lipids. The densities (Fig. 1c and Ext Data Fig. 4) are not complete or high enough resolution to conclude the lipid identity. In addition, it would be helpful to indicate contour (sigma) levels when showing the lipid densities.

Similarly, how was the ordered PC1 lipid in Fig. 4 assigned as PC, but not other lipids? Please discuss if other lipids can contribute to this density.

It is interesting that cryo-EM analysis separated several classes within rotational states 1 and 2. However, only the differences between 1a, 2a and 3 are presented. Can the authors describe the distinct features of all classes?

In all 10 classes representing different rotational states, what are the F1 conformations? Do they represent the same functional/nucleotide state of F1?

In Fig. 3b, "su- β " label is difficult to see. Consider moving the label?

Please show local resolution maps for all final cryo-EM maps. To demonstrate sufficiently high resolution to visualize ordered water molecule around proton channel, FSC curves focusing on relevant regions should be helpful.

Please clarify exactly which gel filtration fractions in Ext Data Fig. 1a were pooled together for cryo-EM study.

Reviewer #2 (Remarks to the Author):

The manuscript by Gahura et al., describes the structure of the ATP synthase from *T. Brucei* as determined by cryo-EM. They identified 10 rotational states that were assembled into a rotational mechanism. They identified localized water molecules in the lumen half-channel, and proximal to the key Arginine residue of subunit a. They discovered structures of subunits b, e, and g that were different from other species. In particular e and g were a key part of an oligomerization interface that was different from other studied ATP synthase from various species. RNAi knockdowns revealed that disrupting the interface affected dimerization and cristae formation in the mitochondria, while the enzyme retained some activity.

This is an important and interesting paper regarding the role of the ATP synthase in cristae formation, and its place in evolutionary schemes. It also provides further insights into mechanisms of ATP synthesis.

There were several features of the structure that were not clearly explained or illustrated:

1. Subunit b is found in all ATP synthases (while being rather divergent), and was discussed here

(para 3 of the Results). But no image illustrates the differences with other b subunits: the N-terminal transmembrane domains and the shortened C-terminus.

2. The unusual beta-barrel of the ring of subunits c was illustrated, but not discussed very much. Apparently it is the N-terminus, but not clear if it is the entire N-terminus, or is there another segment not resolved? What species have this feature? It would seem to create a significant cavity. The authors speculated that subunit e, C-terminus, interacted there. (But for what purpose?)

3. The rotary mechanism was discussed on p.5 of the Results, but was not entirely clear to me. The angles of 117, 136, and 107 would appear to be related to the positions of gamma relative to alpha/beta. The description of c-ring station relative to F1 does not seem complete

Reviewer #3 (Remarks to the Author):

The studies in the manuscript can be divided into 2 related, but distinct areas. In the first part, the studies investigate the structure of the dimer form of the ATP synthase from *Trypanosoma brucei*. In the second part, the studies concentrate on the role of subunits e and g on the formation of the dimer and their role in forming an active enzyme. The first part reports on the high-resolution structure of the dimer form of the ATP synthase – which consists of 25 subunits and 36 lipid molecules. They also report on the presence of 5 ordered water molecules which likely participate in the proton conduction pathway in *F_o*. The dimerization interface is formed by subunits e and g. The authors report on 10 distinct rotamer structures of the enzyme with a resolution from 3.5-4.3Å resolution. Overall, these structures, and are very good and add to the understanding of the structure, function, and evolution of the enzyme.

The second part of the study looks more specifically at the roles of subunits e and g in the dimerization interface. Here a number of biochemical studies are used to investigate their role in activity and dimerization. The studies also revealed that dimerization is critical when it is the primary source of cellular ATP.

The results in this study are clear and unambiguous. The structures are very good and provide some new insights into the structure and function of the ATP synthase and the mechanism of ATP synthesis.

There is just one suggestion. The authors identify cryo density in the cavity of the c-ring as UTP. I understand the rationale for the assignment, but the density does not seem to be good enough to make the assignment. It would have been good to do analysis on the sample to determine if UTP is bound to the enzyme. But I understand that the assignment is made with an understanding that the identity is not certain.

Gahura et al. Response to Reviewer comments

We warmly thank the Reviewers for their constructive feedback, which has helped to significantly improve the manuscript. Before responding to the Reviewer comments below, we would like to briefly outline the major additions and changes to our revised manuscript. We included a more detailed description of the rotary cycle, and extended Figures 2, ED3 and ED5 with additional panels, as requested. We also performed mass-spectrometry analysis to identify the pyrimidine nucleotide residing in the cavity of the c_{10} -ring that has been modeled based on the density, but unfortunately, the compound hasn't been experimentally detected. All panels containing bar graphs were modified and now show all individual data points, to meet the journal's requirements.

Reviewer #1 (Remarks to the Author):

To generate mitochondrial inner membrane curvature and maintain high level of ATP production, the mitochondrial ATP synthase forms dimers which further arrange into oligomers. In this work, Gahura et al. determined the cryo-EM structure of the ATP synthase dimer from *Trypanosoma brucei*, identified both protein and lipid components, revealed a set of rotational states, and proposed subunit-e/g module as an ancestral oligomerization motif of ATP synthases. The functional importance of subunit-g was supported by their RNAi knockdown, mitochondrial ultrastructure, membrane polarization measurement, and ATP production assay. This manuscript is well written with clearly presented figures, and most results are properly interpreted. The results are interesting to the community of ATP synthase research as well as broader membrane protein biology.

I have a few comments and questions listed below.

For the assignment of cardiolipin molecules, particularly those in Fig. 1c, the author should discuss the possibility of other lipids. The densities (Fig. 1c and Ext Data Fig. 4) are not complete or high enough resolution to conclude the lipid identity. In addition, it would be helpful to indicate contour (sigma) levels when showing the lipid densities. Similarly, how was the ordered PC1 lipid in Fig. 4 assigned as PC, but not other lipids? Please discuss if other lipids can contribute to this density.

- This information has now been added to Fig ED5 and Methods section on model building. Particularly, cardiolipins were assigned based on the presence of a characteristic elongated density branched on both termini, corresponding to two phosphatidyl groups linked by a central glycerol bridge. The assignment of the negative phosphate groups is supported by coordination of positively charged residues, mostly arginines. Monophosphatidyl lipids can be distinguished based on the shape of their headgroups. In PC, the three methyl groups of choline moiety generate a distinct tetrahedral density, which is not observed in the case of amine in PE. None of the headgroup

densities is compatible with other monophosphatidyl lipids possibly present in the inner mitochondrial membrane (phosphatidylserine, phosphatidylinositols).

It is interesting that cryo-EM analysis separated several classes within rotational states 1 and 2. However, only the differences between 1a, 2a and 3 are presented. Can the authors describe the distinct features of all classes?

- We thank the reviewer for the opportunity to discuss our data more extensively, and now added the additional information on page 6 with a reference to Sobti et al., 2020, Nat Comm. In the original version of our manuscript we focused on the most populated main rotational state 1 (5 classes/substates), in which we described (i) counter-directional torque of the $\alpha_3\beta_3$ hexamer (Fig. 2c,d), (ii) bending of F1 (Fig. 2e), and (iii) twisting of the peripheral stalk (Fig. 2f). We reason that the motions observed in the main state 1 likely occur also in the remaining states, but could not be observed due to the lower number of detectable substeps (four and one substeps for the states 2 and 3, respectively). Unequal representation of individual main steps is most likely caused by a symmetry mismatch between the decameric c-ring and the $\alpha_3\beta_3$ hexamer (tenfold vs threefold symmetry), as revealed in bacterial ATP synthase (Sobti et al., 2020, Nat Comm), and observed also in *Polytomella* (Murphy et al., 2019, Science).

In all 10 classes representing different rotational states, what are the F1 conformations? Do they represent the same functional/nucleotide state of F1?

- We added a clarification on page 6. The conformation of F₁ in all 10 classes corresponds to the catalytic dwell, in which the three $\alpha\beta$ dimers attain typical tight closed (TP), loose closed (DP) and open (E) conformation, alternating during transitions between individual rotational states. The catalytic interfaces in TP, DP and E conformations bind ATP, ADP, and no nucleotides, consistently with previous studies in similar settings, including the *T. brucei* F1 crystal structure. As no fresh nucleotides were added shortly before freezing, this is the expected functional state of the enzyme.

In Fig. 3b, “su- β ” label is difficult to see. Consider moving the label?

- The label was highlighted, thank you.

Please show local resolution maps for all final cryo-EM maps. To demonstrate sufficiently high resolution to visualize ordered water molecule around proton channel, FSC curves focusing on relevant regions should be helpful.

- We have added new Extended Data Fig. 3, showing all local resolution maps (which are based on FSC curves of overlapping spheres). The highest resolution of 2.55 Å was observed in the membrane region that includes the half channel with the ordered water molecules. To illustrate that the modeled water molecules are part of the highest resolved region, the new figure includes a

closeup view of the lumenal proton channel. The information about the local resolution was also added on page 8.

Please clarify exactly which gel filtration fractions in Ext Data Fig. 1a were pooled together for cryo-EM study.

- The figure was modified as requested, thank you.

Reviewer #2 (Remarks to the Author):

The manuscript by Gahura et al., describes the structure of the ATP synthase from *T. Brucei* as determined by cryo-EM. They identified 10 rotational states that were assembled into a rotational mechanism. They identified localized water molecules in the lumen half-channel, and proximal to the key Arginine residue of subunit a. They discovered structures of subunits b, e, and g that were different from other species. In particular e and g were a key part of an oligomerization interface that was different from other studied ATP synthase from various species. RNAi knockdowns revealed that disrupting the interface affected dimerization and cristae formation in the mitochondria, while the enzyme retained some activity.

This is an important and interesting paper regarding the role of the ATP synthase in cristae formation, and its place in evolutionary schemes. It also provides further insights into mechanisms of ATP synthesis.

There were several features of the structure that were not clearly explained or illustrated:

1. Subunit b is found in all ATP synthases (while being rather divergent), and was discussed here (para 3 of the Results). But no image illustrates the differences with other b subunits: the N-terminal transmembrane domains and the shortened C-terminus.

- We added the requested illustration in Fig 2c. Particularly, to illustrate the divergence of subunit-*b*, we added a schematic figure comparing the arrangement of transmembrane helices of subunit-*b* and adjacent structural elements in *T. brucei* with another type IV ATP synthase of *Euglena* and the type I ATP synthase represented by *S. cerevisiae*. Accordingly, we also elaborated on the conservancy of subunit-*b* transmembrane helices and their context in different types of ATP synthases in the Discussion section.

2. The unusual beta-barrel of the ring of subunits c was illustrated, but not discussed very much. Apparently it is the N-terminus, but not clear if it is the entire N-terminus, or is there another segment not resolved? What species have this feature? It would seem to create a significant cavity. The authors speculated that subunit e, C-terminus, interacted there. (But for what purpose?)

- We added an expansion in the text with a reference to Pinke et al., 2020. Our mass-spec data indicate that only a single N-terminal residue of mature subunit-*c* has not been built in our model, which is likely due to flexibility. Thus, there is no unresolved segment and the c_{10} -ring beta-barrel is constituted by N-termini of the ten copies of subunit-*c*. The same beta-barrel has been so far reported in the structure of type IV ATP synthase from *Euglena* (Mühleip et al., 2019, *eLife*). Sequence comparison and secondary structure prediction upstream of the highly conserved first transmembrane helix of subunit-*c* from species across eukaryotes suggest that the N-terminal beta-barrel is not found outside of Euglenozoa. However, the analysis has limitations, because the presence of mitochondrial targeting sequences prevents reliable prediction of mature N-termini. Therefore, we added a conservative statement “*The β -barrel has been previously reported also in other type IV ATP synthase from E. gracilis (Mühleip et al., 2019)*” on page 3.

The function of the interaction of subunit-*e* with the beta-barrel or its lipid plug is unclear from our structure. A model, in which retraction of subunit-*e* upon calcium exposure pulls out the lipid plug, which induces disassembly of the *c*-ring and triggers permeability transition pore (PTP) opening, has been proposed (Pinke et al., 2020). We added this information to the revised manuscript. However, because cyclophilin D, the key component of mitochondrial permeability transition, has not been found in *T. brucei* (Bustos et al., 2017, PMID 28933785) a putative function remains unclear in our study.

3. The rotary mechanism was discussed on p.5 of the Results, but was not entirely clear to me. The angles of 117, 136, and 107 would appear to be related to the positions of gamma relative to alpha/beta. The description of *c*-ring station relative to F₁ does not seem complete

- To describe rotary motion, the rotor (the $c_{10},\gamma\delta\epsilon$ subcomplex) is viewed as a rigid body rotating against static F₀. The angles are the rotor step size between classes 1a, 2a and 3 relative to the static F₀ subunit-*a*. We specified this information in the revised manuscript on page 6. We also expanded the text describing the F₁ movement relative to c_{10} -ring throughout the rotary cycle, and complementary information is found in the corresponding supplementary movie.

Reviewer #3 (Remarks to the Author):

The studies in the manuscript can be divided into 2 related, but distinct areas. In the first part, the studies investigate the structure of the dimer form of the ATP synthase from *Trypanosoma brucei*. In the second part, the studies concentrate on the role of subunits *e* and *g* on the formation of the dimer and their role in forming an active enzyme. The first part reports on the high-resolution structure of the dimer form of the ATP synthase – which consists of 25 subunits and 36 lipid molecules. They also report on the presence of 5 ordered water molecules which likely participate in the proton conduction pathway in F₀. The dimerization interface is formed by subunits *e* and *g*. The authors report on 10 distinct rotamer structures of the enzyme with a resolution from 3.5-4.3Å

resolution. Overall, these structures, and are very good and add to the understanding of the structure, function, and evolution of the enzyme.

The second part of the study looks more specifically at the roles of subunits e and g in the dimerization interface. Here a number of biochemical studies are used to investigate their role in activity and dimerization. The studies also revealed that dimerization is critical when it is the primary source of cellular ATP.

The results in this study are clear and unambiguous. The structures are very good and provide some new insights into the structure and function of the ATP synthase and the mechanism of ATP synthesis.

There is just one suggestion. The authors identify cryo density in the cavity of the c-ring as UTP. I understand the rationale for the assignment, but the density does not seem to be good enough to make the assignment. But I understand that the assignment is made with an understanding that the identity is not certain.

- Thank you. We made a dedicated large-scale preparation of the ATP synthase from the parasites, as well as a control sample from bacterial cells with the aim of the compound detection by mass-spectrometry. The analysis was performed at the Swedish Metabolomics Centre in Umeå University, on UHPLC-QqQ MRM mode for highest sensitivity method set to detect possible nucleotides. 330 ul of the sample was used, which is more than the total amount required for cryo-EM structure determination. Unfortunately, no signal has been detected.

We thus added in the text “assigned as UTP, although not experimentally detected.”

Samples can not be separated from the analytical blank (MeOH/H₂O injected on the system).
→ We are not able to detect UTP in any of your samples.

UTP peak should be present here

REVIEWERS' COMMENTS

Reviewer #1 (Remarks to the Author):

The authors have adequately addressed my comments. I support the publication of this manuscript in Nature Communications.

Reviewer #2 (Remarks to the Author):

No further comments. Thank you

Reviewer #3 (Remarks to the Author):

This is a revised version of the manuscript with the same title. The authors have addressed well each of the comments in the review. Overall, this is a very solid study that brings novel and significant information on the structure, function, and evolution of the ATP synthase. This will have appeal to a broad audience.

REVIEWERS' COMMENTS

Reviewer #1 (Remarks to the Author):

The authors have adequately addressed my comments. I support the publication of this manuscript in Nature Communications.

Reviewer #2 (Remarks to the Author):

No further comments. Thank you

Reviewer #3 (Remarks to the Author):

This is a revised version of the manuscript with the same title. The authors have addressed well each of the comments in the review. Overall, this is a very solid study that brings novel and significant information on the structure, function, and evolution of the ATP synthase. This will have appeal to a broad audience.